# Hormonal Blood Pressure Regulation during General Anesthesia Using a Standardized Propofol Dosage in Children and Adolescents Seems Not to Be Affected by Body Weight

**DOI:** 10.3390/jcm9072129

**Published:** 2020-07-06

**Authors:** Gunther Hempel, Anne-Marie Maier, Tobias Piegeler, Sebastian N. Stehr, Jürgen Kratzsch, Claudia Höhne

**Affiliations:** 1Department of Anesthesiology and Intensive Care, University of Leipzig Medical Center, 04103 Leipzig, Germany; anne-m.maier@gmx.de (A.-M.M.); tobias.piegeler@medizin.uni-leipzig.de (T.P.); sebastian.stehr@medizin.uni-leipzig.de (S.N.S.); 2Department of Pediatric Cardiology, Leipzig Heart Center, 04289 Leipzig, Germany; 3Institute of Laboratory Medicine, Clinical Chemistry and Molecular Diagnostics, University of Leipzig Medical Center, 04103 Leipzig, Germany; juergen.kratzsch@medizin.uni-leipzig.de; 4Department of Anesthesiology, Intensive Care Medicine and Pain Therapy, DRK Hospital Berlin-Koepenick, 12559 Berlin, Germany

**Keywords:** pediatric anesthesia, obesity, hypotension, RAAS

## Abstract

Obesity in pediatric surgical patients is a challenge for the anesthesiologist. Despite potentially beneficial properties, propofol might also induce hypotension. This study examined whether a dose adjustment in overweight children could avoid hypotension and if there would be differences regarding hormonal regulation in children under anesthesia. Fifty-nine children undergoing surgery under general anesthesia were enrolled in this prospective observational trial. Participants were allocated into two groups according to their BMI. The induction of anesthesia was conducted using propofol (“overweight”: 2 mg/kgBW, “regular”: 3.2 mg/kgBW). The maintenance of anesthesia was conducted as total intravenous anesthesia. Hormone levels of renin, angiotensin II, aldosterone, copeptin, norepinephrine and epinephrine were assessed at different timepoints. Blood pressure dropped after the administration of propofol in both groups, with a nadir 2 min after administration—but without a significant difference in the strength of reduction between the two groups. As a reaction, an increase in the plasma levels of renin, angiotensin and aldosterone was observed, while levels of epinephrine, norepinephrine and copeptin dropped. By adjusting the propofol dosage in overweight children, the rate of preincision hypotension could be reduced to the level of normal-weight patients with a non-modified propofol dose. The hormonal counter regulation was comparable in both groups. The release of catecholamines and copeptin as an indicator of arginine vasopressin seemed to be inhibited by propofol.

## 1. Introduction

Total intravenous anesthesia (TIVA) is one of the standard regimens for general anesthesia in the pediatric and adolescent population [1,2]. Propofol is usually the agent of choice for the induction and maintenance of anesthesia due to its favorable pharmaco-dynamic and -kinetic properties, which, for example lead to an improved comfort at induction and a reduced incidence of postoperative nausea and vomiting (PONV) at the same time [3,4,5]. However, despite these potentially beneficial properties, propofol might also induce hypotension—defined as a reduction in systolic and/or mean arterial blood pressure by 20%—during or after the induction of anesthesia [6,7]. In a large prospective observational study (APRICOT-trial), the rate of serious perioperative events in pediatric anesthesia was 5.2% [8]. Besides respiratory incidents (3.1%), cardiovascular instability (1.9%) accounted for the largest share of these events. Cardiovascular instability was again due to hypotension in over 50% of cases [8].

There is still no consensus regarding the optimal dose of propofol for induction in pediatric anesthesia due to the fact that several dose-finding studies have examined different primary endpoints [9,10,11]. In overweight or obese children or adolescents, optimal dosing might be even more difficult, as the incidence of hypotension under general anesthesia in these patients might be higher than in the general population [12,13]. It seems possible that this observation might be due to a reduced compensatory increase in heart rate, after the reduction in cardiac afterload induced by the anesthetic agent [14], which might at least in part be explicable by the inhibition of the release of circulating catecholamines [15].

The renin–angiotensin–aldosterone system (RAAS) has a broad impact on blood pressure regulation and hypertension in the pediatric population [16]. It was also demonstrated previously that a specific RAAS localized in adipose tissue might be crucial for the development of hypertension in this particular population [16] and that there might be a positive correlation between obesity and increased plasma levels of renin, as well as of aldosterone [17].

Besides the RAAS, a crucial role for the sympathetic nervous system in the regulation of blood pressure in obese patients has been postulated as well, a higher body mass index (BMI) was shown to be associated with increased levels of norepinephrine [18], potentially leading to a vascular hyposensitivity against norepinephrine and angiotensin II induced by morbid obesity [19].

However, the exact regulatory mechanisms of these hormonal imbalances in children are still unknown, especially in the context of general anesthesia. The current study therefore aimed at evaluating the effect of a propofol-based general anesthesia on hormonal homeostasis in children. Additionally, it should be examined whether a dose adjustment of propofol might lead to a reduction in the incidence of hypotensive events in overweight or obese children undergoing general anesthesia.

## 2. Methods

After approval by the local ethics committee at the University of Leipzig (Protocol-No. 078-12-05032012) and with respect to the World Medical Association Declaration of Helsinki, a total of 59 children between 6 and 16 years old (American Society of Anesthesiologists (ASA) classifications I or II), undergoing a surgical or orthopedic procedure under general anesthesia with endotracheal intubation, were enrolled into this prospective observational trial. The consent for enrollment was provided by the child’s parents or legal guardian. The following patients were considered not to be eligible for this study: patients who might have been pregnant or where no consent could be obtained, patients with malignant diseases or under long-term medication with corticosteroids, opioids or any other sedative agent. The study was registered in the German Clinical Trials Register (DRKS00021667, retrospective registration).

Due to the observational nature of the study, there was no randomization. However, participants were allocated into two different observational groups according to their age- and gender-specific BMI: Children would be classified as “overweight”, if their BMI was higher than (or equal to) the 90th percentile, all other children were considered to be of “regular weight”, i.e., below the 90th percentile. The distribution of the groups is shown in Figure 1.

### 2.1. Induction and Maintenance of Anesthesia

In accordance with the current guidelines of the European Society of Anesthesiology, all patients were allowed to drink clear liquids up to two hours before the induction of anesthesia [20]. An oral premedication with midazolam 0.4 mg/kg body weight (BW) (maximum of 10 mg) was given prior to the transfer to the pre-operative holding area. The induction of anesthesia was conducted via the intravenous administration of fentanyl (2 µg/kgBW), propofol (2 mg/kgBW for “overweight” children, 3.2 mg/kg for “regular weight” children) [11] and cis-atracurium (0.1 mg/kgBW) for muscle paralysis. If necessary, the anesthesiologist in charge was allowed to administer additional boluses of propofol in 0.5 mg/kgBW increments. In all patients, the maintenance of anesthesia was conducted by a continuous intravenous administration of propofol via an infusion pump at an initial rate of 10 mg/kgBW/h, which could be reduced during the course of the anesthesia depending on hemodynamic stability, as well as on the monitored bispectral index (BIS). BIS target values were 30 to 40. Fentanyl was administered repeatedly for sufficient analgesia as a bolus of 1–2 µg/kgBW according to clinical criteria. The various operations were then started promptly after release by the anesthesiologist and were carried out independently of the further measurements up to 45 min after the induction of anesthesia.

### 2.2. Monitoring

Blood pressure and heart rate were recorded twice before the induction of anesthesia (first measurement after arrival in the pre-operative holding area, second measurement immediately before the induction of anesthesia), then every two minutes for the first 10 min after propofol administration, and after that every 5 min until 45 min after propofol administration. The size of the blood pressure cuff was adjusted according to the manufacturer’s specifications in each case to the circumference of the upper arm to avoid systematic incorrect measurements.

### 2.3. Measurement of Hormone Levels

Plasma levels of renin, angiotensin II, aldosterone and copeptin, as well as norepinephrine and epinephrine were assessed after arrival in the pre-operative holding area (T0), 15 min (T1) and 45 min after the induction of anesthesia (T2). The amount of blood taken was 5 mL up to a maximum of 10 mL at each measurement point. Blood samples for renin were stored at room temperature, samples for the evaluation of plasma levels of all other hormones were immediately cooled and centrifuged at 4 °C and 3000 rpm for 15 min (Jouan BR 3.11 centrifuge, Thermo Fisher Scientific, Waltham, MA, USA) and then stored at −80 °C until further analysis. The assessment of angiotensin II levels was performed using a radioimmunoassay after sample extraction (DRG, Marburg, Germany). We handled three different test kits with individual reference probes for the recovery of the analyte after extraction. The comparability of the individual extraction procedures was ensured by the adjustment of all samples to 100% recovery. Plasma levels of epinephrine and norepinephrine were examined via liquid chromatography and electrochemical detection, as previously described [21,22].

All other relevant information, like blood pressure and heart rate to the defined time points and the consumption of infusions, propofol, fentanyl and cis-atracurium, as well as the duration of anesthesia and the operation, were recorded in a separate study protocol.

### 2.4. Statistical Analysis

Categorical data are reported as the number of patients, together with the corresponding percentage of the total (“regular weight” versus “overweight”). Differences between the proportions of qualitative data, such as the patients’ gender, ASA status or type of surgery were assessed with the χ^2^ or the Fisher’s exact test, where appropriate. Due to the relatively small sample size, no normal distribution of the values was assumed. Non-parametric data are reported as median (interquartile range (IQR)) and were compared using a Mann–Whitney U test. Blood pressure and hormone levels over time were examined using the Friedman test. A *p*-value of <0.05 was considered to be statistically significant. The statistical analysis and the creation of the figures were performed using SPSS (Version 18.0.1, SPSS Inc., Chicago, IL, USA) and Microsoft Excel (Version 2016, Microsoft Corporation, Redmond, WA, USA).

## 3. Results

### 3.1. Demographics and Anesthesia-Related Parameters

A total of 59 patients were included in the study and underwent further analysis. Table 1 and Table 2 show demographic data for both treatment groups, as well as the most important parameters related to anesthesia.

Patients in the “overweight” group had a significantly higher body weight (median 71 kg (IQR 53.8–80.6) for “overweight” vs. 48.5 kg (33.5–54.5) for “regular weight”, *p* < 0.001) and corresponding BMI (25.8 (23.0–29.4) for “overweight” vs. 18.2 (15.7–20.8) for “regular weight”, *p* < 0.001) than patients in the “regular weight” group. Additionally, percentiles for height were higher in the “overweight” group (87 (40–96) vs. 55 (28–78), *p* = 0.008). There was no significant difference between the groups in terms of the distribution of the ASA status (*p* = 0.368) and the type of surgery (*p* = 0.478).

Patients in the “regular weight” group received a larger amount of crystalloid fluids in relation to body weight (4.5 mL/kgBW (3.2–5.7) for “overweight” vs. 7.4 mL/kgBW (5.4–8.9) for “regular weight”, *p* < 0.001) as well as of propofol (cumulative dose over 45 min; 10.2 mg/kgBW (9.6–11.6) for “overweight” vs. 11.6 mg/kgBW (10.9–12.2) for “regular weight”, *p* = 0.005) compared to patients in the “overweight” group (Table 1 and Table 2).

### 3.2. Blood Pressure

A Spearman’s correlation analysis examining the relationship between the preoperative systolic blood pressure and the patients’ BMI revealed a moderate correlation between these two factors (ρ = 0.452, *p* < 0.001, Figure 2). “Overweight” patients had a significantly higher systolic blood pressure during the course of anesthesia compared to “regular weight” patients (except for measurements 6 and 8 min after the induction of anesthesia, Figure 3). Systolic blood pressure dropped in both groups after the administration of propofol, with the nadir observed 2 min after the administration of the drug.

Preoperative systolic blood pressure was also correlated with higher plasma levels of both aldosterone (ρ = 0.274, *p* = 0.039), as well as of norepinephrine (ρ = 0.485, *p* < 0.001). All other hormone levels did not show any significant relationships, except for a correlation between plasma renin levels with the patients’ BMI (ρ = 0.363, *p* = 0.005; all Table 3 and Table 4).

### 3.3. Changes in Hormone Levels During the Course of Anesthesia

Focusing on the hormone levels at the three different time points (preoperative (T0), 15 min and 45 min after the induction of anesthesia (T1 and T2)), there were significant changes in all measured parameters. The hormones of RAAS (renin, angiotensin II and aldosterone) showed an increase in plasma levels over the course of time, while epinephrine and norepinephrine levels dropped. In the comparison of the groups (regular weight vs. overweight), there was only a difference in epinephrine at timepoint T2, whereby the level in the overweight patients decreased to be significantly less in absolute terms compared to the initial value (*p* = 0.031). Figure 4 shows an overview of all hormone levels in the course of the measuring times. Unfortunately, not all hormone levels could always be measured for all patients. The reason for this was, among others, insufficient amounts of blood taken afterwards, as well as instabilities in the performance of the measurements in the laboratory itself. In two children of the regular weight group, the blood samples could not be analyzed at all. In the normal weight group, 39–40 of the patient blood samples could be analyzed correctly at the different measurement points regarding the RAAS/copeptin and 31–37 of the samples regarding the catecholamines. In the overweight group, 16–17 of the patient blood samples could be analyzed correctly at the different measurement points regarding the RAAS/copeptin and 14–15 of the samples regarding the catecholamines (see Appendix A).

### 3.4. Limitations

A limitation of the current study could be the monocentric design with a relatively small number of patients. The number of patients is partly due to the rather complex laboratory measurements but is comparable to other similar studies. Furthermore, due to the study design and the research question, it was not possible to randomize the patients or to blind the investigators. Due to the lack of comparable data at the time of the study design, no power analysis/sample size calculation was performed in advance. However, an exemplary post hoc power analysis for the correlation of BMI with blood pressure (see Figure 2) resulted in a value of 0.957 (for *n* = 59 with α = 0.05 and ρ = 0.452), which corresponds to a high statistical power.

## 4. Discussion

We found a moderate positive correlation between patients’ BMI and preoperative systolic blood pressure (ρ = 0.452, *p* < 0.001). This result is in accordance with previous investigations, which in the meantime have been able to prove a corresponding connection [23,24,25]. In addition to the BMI, other factors such as the hip/waist circumference or body shape index are increasingly being examined, which also show a positive correlation and a cardiovascular risk [26,27]. Many of the well-known consequences of high blood pressure, such as stroke and myocardial infarction, are rarely seen in childhood, but it could lead to hypertension in adults. The overweight patients in our study had a significantly higher systolic blood pressure during the course of anesthesia (except for measurements 6 and 8 min after induction of anesthesia). Just like in adults, there are no clear guidelines on how anesthesia should be administered in arterial hypertension [28]. It seems to make sense from a pathophysiological point of view not to let the mean arterial pressure drop to values lower than 25–30% of those of the waking state. Noninvasive cardiac output monitoring and near-infrared spectroscopy (NIRS) should be considered to ensure good end organ perfusion [28].

This is especially true since obese children seem to be prone to relevant hypotension after anesthetic induction (preincision hypotension) [6,12,24]. By adjusting the dose of propofol to induce anesthesia in overweight and obese patients, no significant difference in the proportion of hypotension was found between the two groups [11]. Although propofol could be titrated after the initial dose in both groups according to clinical need, the patients of the “overweight group” received significantly less propofol per kilogram of body weight than the normal weight group 10 min after anesthetic induction. Since it could not be shown that overweight children show relevant hypotension more frequently, the results of the present study suggest that the reduced introductory dose of propofol with titrated additional dosage according to clinical requirements, especially in the risk group “overweight”, might lead to a significant reduction in intraoperative hypotension.

Looking at the amount of crystalloids applied in the period under investigation, there is also a significant difference between the two groups. In relation to body weight, the patients in the “overweight” group received significantly less fluid per kg of bodyweight. However, the absolute amount in mL did not differ significantly between the groups and was a median of 350 mL in both. It can only be assumed that the clinical routine of the anesthesiologists was more important than a weight-adapted procedure.

Regarding hormonal regulation, three essential aspects stand out: After the induction of anesthesia, there was a continuous increase in the components of the RAAS (renin, angiotensin and aldosterone). However, no significant increase in the release of copeptin, as a surrogate parameter of arginine vasopressin (AVP), with a decrease 15 min after anesthesia induction and a return to approximately baseline levels after 45 min, was observed [29]. The catecholamines showed a significant decrease after anesthetic induction in both groups, with hormone levels remaining below the baseline level throughout the measurement period.

The lack of counter regulation of the catecholamines epinephrine and norepinephrine can most likely be explained by a propofol-induced blockade of the sympathetic nervous system [15,30,31]. The only significant difference between the two groups was in the epinephrine level 15 min after the induction of anesthesia, whereas the overweight group showed significantly lower values (*p* = 0.005). The basic course of hormone levels did not differ between the two groups. Copeptin levels also showed no significant increase, although their release would have to be induced by a relative lack of volume. A possible explanation might be that the release of copeptin (and thus the effect of AVP) could be inhibited by propofol as well [32]. This effect also appears to be independent of a patient’s weight, as there was no significant difference between the two groups.

The RAAS therefore appears to make the essential contribution to maintaining and restoring adequate blood pressure levels in both groups. All components rise equally during the course of anesthesia, whereby the initial values did not differ significantly between the two groups and, at the later measuring points, no significant difference could be detected for one of the hormones. This therefore indicates that this form of hormonal regulation is independent of BMI.

The impact of fentanyl on hormonal regulation is not yet fully understood. It seems logical that untreated intraoperative pain causes an increase in various stress hormones (e.g., catecholamines). Therefore, as in any anesthesia, sufficient pain therapy was given a high priority. During the study, there was no significant difference in the consumption of fentanyl between the two groups based on body weight. It was concluded in older studies, with high doses of opioids, that the suppression of hemodynamic or hormonal stress responses is not related to opioid dose [33]. Apart from that, the RAAS itself seems to have a role in modulating pain, and therefore might interact with fentanyl [34]. Concerning the copeptin concentrations, it seems possible that fentanyl or some fentanyl-mediated effects may have an influence. An experimental study with 16 healthy adults found a strong association between fentanyl and copeptin concentrations [35].

## 5. Conclusions

Taking various influencing factors into account, the current study shows that the increased rate of preincision hypotension in overweight children can be reduced by adjusting the propofol dosage to the level of normal weight patients with a non-modified propofol dose. Nevertheless, in both groups, relevant drops in blood pressure occur after the induction of anesthesia with propofol. The hormonal counter regulation seems to take place in both groups in a comparable way, primarily through the activation of the components of the RAAS. The release of catecholamines and copeptin, as an indicator of AVP, seems to be inhibited by propofol as well. Further investigations in larger study populations appear necessary to prove our findings and to clarify the influence of other hormones on the blood pressure regulation of normal and overweight children under general anesthesia.

## Figures and Tables

**Figure 1 jcm-09-02129-f001:**
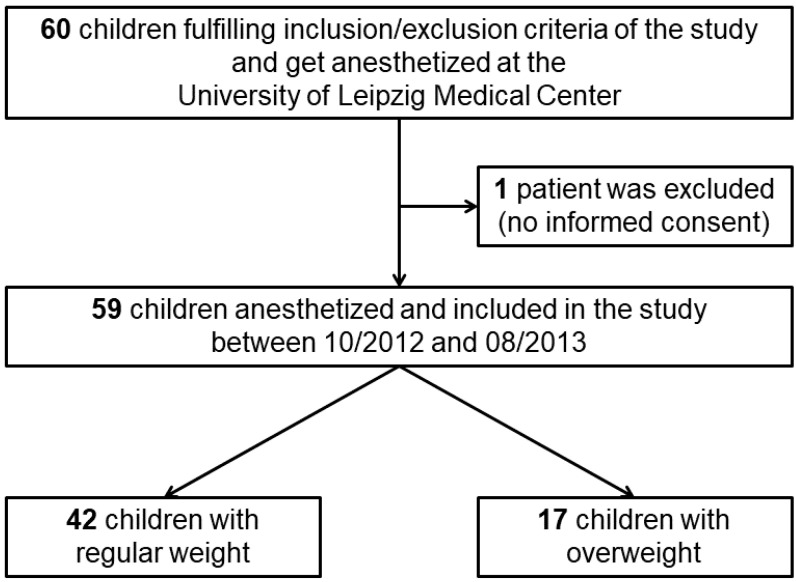
Flowchart of study enrollment.

**Figure 2 jcm-09-02129-f002:**
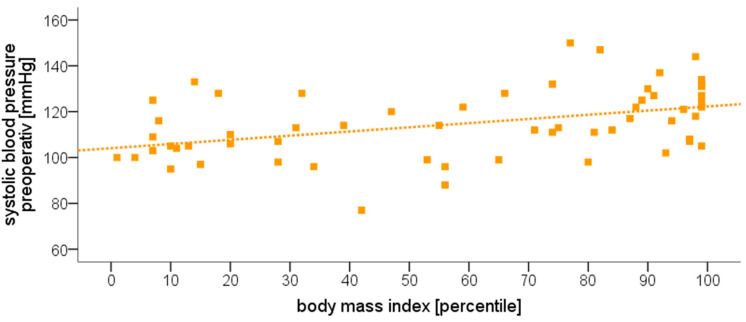
Correlation between body mass index and preoperative systolic blood pressure for all patients (*n* = 59; Spearman’s correlation: ρ = 0.452, *p* < 0.001). The squares stand for individual patients, the dotted line symbolizes the correlation.

**Figure 3 jcm-09-02129-f003:**
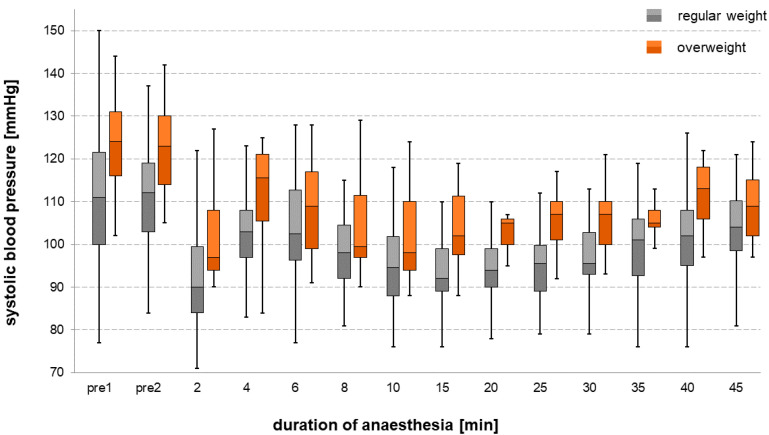
Systolic blood pressure over time during the course of anesthesia comparing the two groups “regular weight” and “overweight” represented in boxplots. The patients in the overweight group had a significantly higher systolic blood pressure during the course of anesthesia (except for measurements 6 and 8 min after the induction of anesthesia), with a drop in both groups 2 min after the administration of propofol. Pre1 and pre2 stand for blood pressure measurements before anesthetic induction—the minutes to the right of these stand for the time span from the initial administration of propofol.

**Figure 4 jcm-09-02129-f004:**
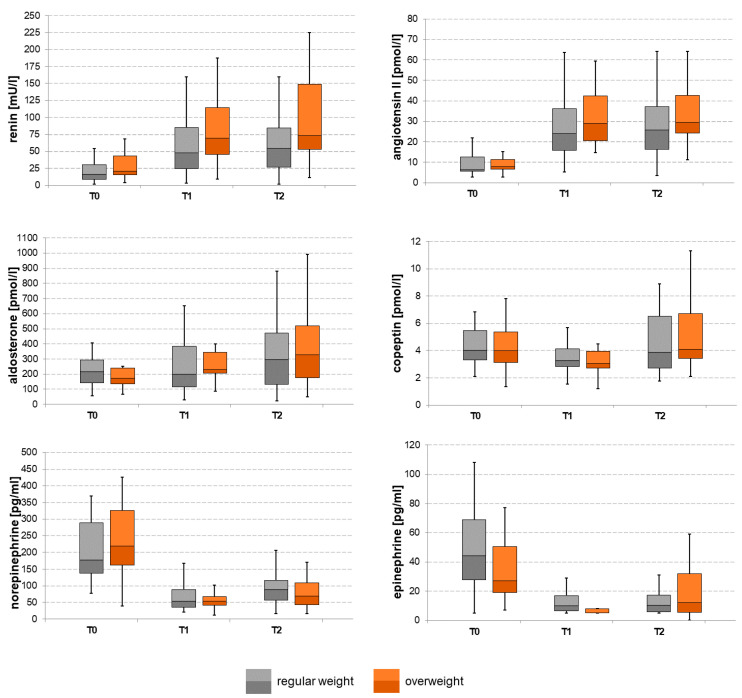
Hormone levels of renin, angiotensin II, aldosterone, copeptin, norepinephrine and epinephrine during the course of anesthesia comparing the two groups “regular weight” and “overweight” represented in boxplots. The hormone levels were measured at three different time points: preoperative (T0), 15 min (T1) and 45 min after the induction of anesthesia (T2). In the comparison of the groups, there was only a difference in epinephrine at timepoint T2, whereby the level in the overweight patients decreased to be significantly less in absolute terms compared to the initial value (*p* = 0.031).

**Table 1 jcm-09-02129-t001:** Demographic data (median (interquartile range (IQR)); (1) Mann–Whitney U test; (2) Fisher’s exact test; BMI: body mass index).

		Group “Regular Weight”	Group “Overweight”	*p*-Value (1)
n		42	17	
Sex	Male (%)	19 (45.2%)	11 (64.7%)	0.252(2)
	Female (%)	23 (54.8%)	6 (35.3%)	
Age (years)		12.5 (10–14)	12 (9–14)	0.775
Height (cm)		158 (141–165)	164 (149–171)	0.162
	percentile	55 (28–78)	87 (40–96)	0.008
Weight (kg)		48.5 (33.5–54.5)	71.0 (53.8–80.6)	<0.001
	percentile	45.5 (25–70)	98 (95–99)	<0.001
BMI (kg/m^2^)		18.2 (15.7–20.8)	25.8 (23.0–29.4)	<0.001
	percentile	41 (14–74)	98 (94–99)	<0.001
ASA (n (%))	1	29 (69.0%)	9 (52.9%)	0.368(2)
	2	13 (31.0%)	8 (47.1%)	
Type of surgery (n (%))	Ortho/trauma surgery	32 (76.2%)	15 (88.2)	0.478(2)
	General surgery	10 (23.8%)	2 (11.8%)	

**Table 2 jcm-09-02129-t002:** Anesthesia-related parameters (median (interquartile range (IQR)); (1) Mann–Whitney U test; BW: body weight).

	Group “Regular Weight”	Group “Overweight”	*p*-Value (1)
Duration of anesthesia (min)	118 (88–162)	116 (103–139)	0.847
Duration of surgery (min)	61 (42–110)	60 (53–79)	0.913
Start of surgery (min after induction of anesthesia)	29.5 (25.8–37.3)	33.0 (28.5–41.0)	0.180
Propofol dosage for induction of anesthesia (mg/kg*BW)	3.2 (3.2–3.3)	2.0 (2.0–2.1)	<0.001
Propofol cumulative up to 45 min (mg/kg*BW)	11.6 (10.9–12.2)	10.2 (9.6–11.6)	0.005
Fentanyl dosage at induction (µg/kg*BW)	2.0 (2.0–2.1)	2.0 (2.0–2.1)	0.966
Fentanyl cumulative up to 45 min (µg/kg*BW)	3.7 (3.0–4.3)	3.2 (2.9–3.9)	0.170
Crystalloid (mL/kg*BW)	7.4 (5.4–8.9)	4.5 (3.2–5.7)	<0.001
Crystalloid cumulative up to 45 min (mL)	350 (200–450)	350 (175–450)	0.781

**Table 3 jcm-09-02129-t003:** Correlation of preoperative systolic blood pressure with different preoperative hormone levels.

Correlation of Preoperative Systolic Blood Pressure with Preoperative Levels of	Spearman’s Rank Correlation Coefficient (ρ)	*p*-Value
Renin (mU/L)	0.158	0.240
Angiotensin II (adj.) (pmol/L)	0.121	0.369
Aldosterone (pmol/L)	0.274	0.039
Copeptin (pmol/L)	−0.193	0.155
Epinephrine (pg/mL)	0.141	0.324
Norepinephrine (pg/mL)	0.485	<0.001

**Table 4 jcm-09-02129-t004:** Correlation of body mass index with different preoperative hormone levels.

Correlation of Body Mass Index with Preoperative Levels of	Spearman’s Rank Correlation Coefficient (ρ)	*p*-Value
Renin (mU/L)	0.363	0.005
Angiotensin II (adj.) (pmol/L)	0.082	0.545
Aldosterone (pmol/L)	0.146	0.277
Copeptin (pmol/L)	−0.226	0.094
Epinephrine (pg/mL)	−0.055	0.701
Norepinephrine (pg/mL)	0.019	0.180

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
