# Peer review of "Hormonal Blood Pressure Regulation during General Anesthesia Using a Standardized Propofol Dosage in Children and Adolescents Seems Not to Be Affected by Body Weight"

_jcm, 2020, doi:10.3390/jcm9072129_

Round 1

Reviewer 1 Report

Thank you for submitting the manuscript. I carefully read the new version of the manuscript, appreciating the changes. The new title more appropriately reflects the actual contents of the manuscript. In the introduction, the reference to the APRICOT study shows that intraoperative hypotension is an important topic in pediatric anesthesia in relation to complications. This concept adds value to the manuscript by emphasizing how the subject matter is important for patient safety- The study limits described also explain what the weaknesses of the manuscript are and how further studies are needed to verify the claims. In fact, the small number of patients and the monocentric nature of the study cannot justify definitive conclusions.

The paragraph on analgesia satisfies the request for clarification regarding how intraoperative analgesia was maintained.

In conclusion, the revisions of the manuscript improving it from a methodological, conceptual and results point of view. The topic of the paper is important and innovative and I appreciated its originality.

I hope these comments are useful.

Kind Regards.

Author Response

Dear Reviewer 1,

Thank you very much for your feedback on the revision of our manuscript and your final positive evaluation.

Kind regards,

Gunther Hempel on behalf of all authors

Reviewer 2 Report

The authors should at least unify the types of surgery, and the other factors (surgical invasion, blood loss, dose of drugs, the total amount of fluid and so on) are naturally be similar.

Author Response

Dear Reviewer 2,

We would like to thank your very much for your feedback and for the additional suggestions.

The authors should at least unify the types of surgery, and the other factors (surgical invasion, blood loss, dose of drugs, the total amount of fluid and so on) are naturally be similar.

As mentioned in the methods section of our manuscript, the anesthetic procedure was performed according to our institutional protocol, using a standard infusion regime and standard medication in both groups. Surgical procedures were short, had an average duration of 60 minutes in both groups (see Table 2) and blood loss was considered to be negligible (therefore not mentioned). Crystalloid infusion dose corrected for body weight was significantly lower in the overweight group. Yet, the absolute amount of infused crystalloids - not corrected for body weight - was not significantly different between the two groups.

In order to highlight the comparability of the groups in the manuscript to a better degree, we now divided Table 1 (now Tables 1 and 2) and added additional information. Table 1 now summarizes the demographic data. Here, we have added the type of surgery. A comparison of the two groups shows no significant difference (p=0.478) between operations in orthopedics/traumatology and general/visceral surgery. Table 2 summarizes anesthesia-related parameters in both groups. We added the absolute amount of infused crystalloids at the end of the table – there was no significant difference between the two groups (p=0.781). The labelling of the tables was adjusted accordingly.

As a consequence of these changes in the tables, we have added a note in line 149 on the statistical calculation of possible differences in the distribution of surgery types. We have added a note on the statistical comparability of the groups regarding the ASA status and the type of surgery in the text from line 166 onwards. In addition, we have added a reference to Table 2 in the brackets in line 173 and at the beginning of the "results" section in line 159.

Finally, we adjusted the labelling of the other tables - so the old table 2 became table 3 and the old table 3 became table 4.

We hope that these changes in our manuscript take your comments into account.

A standardization purely on the basis of the type of operation also seems difficult in our eyes. It can at least be assumed that many formally identical operations on overweight and obese children are more difficult from an operational point of view and therefore take longer.

Kind regards,

Gunther Hempel on behalf of all authors

Reviewer 3 Report

Dear authors,

Thanks for this appropriately reviewed version of your work.

Author Response

Dear Reviewer 3,

Thank you very much for your feedback on the revision of our manuscript and your final positive evaluation.

Kind regards,

Gunther Hempel on behalf of all authors

Round 2

Reviewer 2 Report

The title should include information on adjusting the propofol dosage.

Author Response

Dear reviewer,
thank you for your review and your note regarding the title of the manuscript.
We have now changed the title once again and hope to take sufficient account of the feedback from all reviewers.
The new title of the manuscript is:

Hormonal blood pressure regulation during general anesthesia using a standardized propofol dosage in children and adolescents seems not to be affected by body weight

Kind regards,
Gunther Hempel on behalf of all authors

This manuscript is a resubmission of an earlier submission. The following is a list of the peer review reports and author responses from that submission.

Round 1

Reviewer 1 Report

Dear authors, 

Thanks for submitting your work to the journal. You investigated whether an adjusted propofol dose in overweight children is associated with hypotension and hormones seric levels vs. a fixed dose in non overweight children. You conclude that the blood pressure and the hormonal levels were comparable amongst the groups. 

The work is relevant, well performed and easy to read. 

The reviewer would suggest to:

  • be cautious in the interpretation as two variables are simultaneously changed (BMI and propofol dose). So, you cannot conclude about the effect of the propofol dose adjustment. But this does not preclude to conclude about the effect of a modified dose in overweight children vs. non modified dose in non overweight children. 
  • add a paragraph including a power analysis (post hoc if not done a priory, but permitting to convince the reader that a clinically significant difference was possible to detect vs. to discuss an underpowering as appropriate.
  • explain how the blood pressure cuff was chosen, to reassure the reader about a systematic bias linked to an undersized cuff. 

Reviewer 2 Report

The authors studied the effect and its mechanism of propofol on blood pressure, RAAS hormones, copeptine and catecholamines in obese children and adolescents by adjusting the propofol dosage. The results showed that propofol induced hypotension, an increase in RAAS hormones and a decrease in copeptine and catecholamines without a significant difference between the regular and overweight groups. They concluded that adjusting the propofol dosage can relieve intraoperative hypotension in overweight children to the level of regular weight children.

There are several important issues regarding the interpretation of these findings.

Major comments

  1. The authors concluded that the increased rate of preincision hypotension in overweight children can be reduced to the level of normal weight patients by adjusting the propofol dosage. That is misleading. In the current study, the anesthesiologists kept blood pressure adequate in the both groups by adjusting the propofol and fentanyl dosage and infusion of crystalloid at their direction. There are too many factors that affect blood pressure. If the regular weight children received the same amount of crystalloid as that of the overweight children, the regular weight children may have received the same amount of propofol as that of the overweight children.
  2. The authors should discuss the effect of fentanyl on the regulation of the RAAS, copeptine and catecholamines.
  3. The title does not reflect the content of the manuscript. If the obese children require less propofol to keep adequate blood pressure than the normal weight children, obesity rather affects hormonal regulation of blood pressure during general anesthesia in children and adolescents.
  4. The methods section
  • Does ‘preincision hypotension’ mean that the starts of surgery were stopped until 45minutes after induction of anesthesia?
  • How much is the amount of extracted blood of one patient for the measurement of hormones and catecholamines?
  • What is the statistics software used in this study?
  1. The authors should add figure legends. The figure and its legend must be understandable without reference to the text. Include definitions of any symbols used and define/explain all abbreviations and units of measurement.

Minor comments

  1. Copeptine and copeptin coexist in the text.

Reviewer 3 Report

Thank you for submitting your manuscript.

I read with great interest the manuscript, which is well written and original in content. I suggest two minor revisions. The first suggestion is to improve the introduction by better explaining the cardiovascular risk in pediatric anesthesia and I suggest one reference that deal with the topic : Incidence of Severe Critical Events in Paediatric Anaesthesia (APRICOT): A Prospective Multicentre Observational Study in 261 Hospitals in Europe Lancet Respir Med. 2017 May;5(5):412-425. doi: 10.1016/S2213-2600(17)30116-9. Epub 2017 Mar 28.

The second suggestion is to better clarify how analgesia is guaranteed during the maintenance of anesthesia in section 2.1 (Induction and maintenance of anesthesia).

I hope these comments will be helpful.

Round 2

Reviewer 2 Report

The study protocol is arbitrary. Again, there are too many factors (propofol, fentanyl, crystalloid and surgical invasion) that affect blood pressure. The dose of drugs except for propofol and infusion rate should be fixed in accordance with body weight to investigate the effect of propofol. It is that the anesthesiologists, not propofol that regulated blood pressure in this study.